# Vital Needs of Dutch Homeless Service Users: Responsiveness of Local Services in the Light of Health Equity

**DOI:** 10.3390/ijerph20032546

**Published:** 2023-01-31

**Authors:** Coline van Everdingen, Peter Bob Peerenboom, Koos van der Velden, Philippe Delespaul

**Affiliations:** 1Department of Psychiatry and Neuropsychology, Maastricht University, 6200 MD Maastricht, The Netherlands; 2Van Everdingen Health Care Consultancy, 6132 TP Sittard, The Netherlands; 3Tangram Health Care Consultancy, 7006 CP Doetinchem, The Netherlands; 4Department of Primary and Community Care, Radboud University Medical Centre, 6500 HB Nijmegen, The Netherlands; 5Mondriaan Mental Health Trust, 6401 CX Heerlen, The Netherlands

**Keywords:** public health policy, homelessness, health equity, rights-based ecosystem approach, 3-D recovery

## Abstract

Background: Healthcare and social services aim to ensure health equity for all users. Despite ongoing efforts, marginalized populations remain underserved. The Dutch HOP-TR study intends to expand knowledge on how to enable the recovery of homeless service users. Methods: A naturalistic meta-snowball sampling resulted in a representative sample of homeless services (N = 16) and users (N = 436). Interviews collected health and needs from user and professional perspectives in a comprehensive, rights-based ecosystem strategy. We calculated the responsiveness to needs in four domains (mental health, physical health, paid work, and administration). Results: Most service users were males (81%) with a migration background (52%). In addition to physical (78%) and mental health needs (95%), the low education level (89%) and functional illiteracy (57%) resulted in needs related to paid work and administration support. Most had vital needs in three or four domains (77%). The availability of matching care was extremely low. For users with needs in two domains, met needs ranged from 0.6–13.1%. Combined needs (>2 domains) were hardly met. Conclusions: Previous research demonstrated the interdependent character of health needs. This paper uncovers some causes of health inequity. The systematic failure of local services to meet integrating care needs demonstrates the urgency to expand recovery-oriented implementation strategies with health equity in mind.

## 1. Introduction

Around the world, healthcare and social services intend to offer equal chances of health and growth for all inhabitants. However, proven effective interventions persistently fail to reach marginalized populations. To date, local inclusion health movements need to prove their power to bridge huge health disparities. Therefore, in 2016 the US Institute for Healthcare Improvement published a white paper titled “Achieving health equity” [1]. It presented inspiring examples from practice together with scientific evidence in an economic business case. It provided a clear framework for organizations on how to make health equity a strategic priority and tailor resources and organizational processes to that aim. Accurate data that disclose where and why disparities exist enable a focus on health equity in organizations and communities.

Since then, the plead grows louder to tackle preventable and unjust health inequalities related to the social determinants of health. Recently, the COVID-19 pandemic uncovered the potentially opposite effects of public health strategies in marginalized populations. This heightened social awareness of the preexistent, extreme impact of health inequity fostered discussion about essential healthcare system transformations [2,3,4,5]. The American College of Cardiology argued for the Quintuple Aim case evaluation. It recollected the introduction of Triple Aim in 2007, showing that improved patient experiences, better outcomes, and lower costs go hand in hand. Quadruple Aim adds an important focus on clinicians’ well-being in 2014. Recently, health equity was added resulting in Quintuple Aim in 2021. A viewpoint paper from the American Medical Association mentioned that “quality improvement without equity is a hollow victory” [2]. It addressed that intersectoral data sharing can contribute to making health inequity and its underlying causes manifest. 

Marginalized populations have to face accumulating health and social disadvantages, which can result in homelessness [6,7,8,9,10]. Integrating care models, such as Housing First, proved effective to enable recovery [11,12]. Despite tailored adaptations to suit homeless subpopulations’ needs in local contexts, the outcome and community integration of these integrating models remain low [6,13,14]. Building collaborative services focused on recovery remains challenging [15,16,17]. Consequently, homeless populations remain underserved. They still have to endure the consequences of extreme inequity and injustice [18,19]. A meta-analysis by Aldrigde and colleagues [20] identified a similar, extensive impact of health inequity on mortality and morbidity outcomes in homeless individuals, prisoners, sex workers, and substance users. Cohort research in various countries revealed how the substantive prevalence of mental illness among Homeless Service Users (HSU) contributes to increased mortality rates. Based on the Danish Homelessness Register, Nielsen et al. [21], identified standardized mortality ratios of 6.7 and 5.6 for homeless women and men. The results disclosed a doubled risk of dying among HSU with Severe Mental Illness (SMI) compared to inhabitants with SMI in the general population. The research among Dutch HSU in Rotterdam revealed a rate ratio of 5.6 and 3.3 among homeless women and men [22]. The remaining life expectancy at 20 years decreased to 15.8 (women) and 14.3 years (men). The Public Mental Healthcare Register in Utrecht revealed rate ratios of 2.7 (women) and 3.0 (men) [23].

During the past decade, insight increased as to why inadequate service access causes inequity. Levesque, Harris, and Russell operationalized access as the opportunity for the fulfillment of service needs [24]. The access is the result of a dynamic interaction process between services and users. The users’ needs and behavior are central and considered in relation to demand and supply factors. The demand side focuses on users’ abilities and preferences, while the supply side concentrates on the role of services in shaping recovery-enabling conditions. Building on this framework, the Health Quality and Safety Commission of New Zealand identified the main drivers for quality improvement in light of health equity [5]. Daily life conditions related to income, employment, housing, and social support shape the demand side. On the supply side, consumer engagement and cultural competence are important, in addition to health technology and leadership. Health literacy is an important driver on both sides of the interaction process.

Research in many countries highlighted supply side factors. The PROMO quality review in 14 European capitals shows low levels of outreach and a lack of input from qualified mental health staff [25]. Social network analyses in deprived areas of two capitals revealed that collaboration within organizations was much higher than between organizations [26]. In 2020, Rosen, Gill, and Salvador-Carulla explicitly addressed the failure of healthcare systems to meet complex, integrating care needs, such as in homeless populations [27]. They argued to reframe community healthcare in a healthcare ecosystem approach, targeted at balanced care [28] and rooted in the keystones of community (mental) health (person-centeredness, recovery, human rights, challenging stigma, and discrimination). During the pandemic, scarcity of resources forced service providers to choose between hospital admission for COVID-19 or treatment of cancer. The awareness of health professionals to protect the rights and safety of their patients increased. The International Council of the Patient Ombudsman addressed that the care safety for people with chronic health conditions was at stake [29]. Its founder substantiated why healthcare systems should be able to respond effectively to diverse population needs.

Participatory research highlighted the demand side of access to the care process. For instance, the Canadian Observatory on Homelessness (COH) identified (un)helpful conditions for recovery based on peer narratives [30]. The peer voices revealed the inefficiency of “clinical, oppressive, and institutional help” in the dominant medical symptom reduction approach. By contrast, they argue that reciprocal relations with compassionate helpers are more helpful. They move beyond “the textbook” to realize flexible, adaptive, strength-based care. They underpinned that “homelessness is not an issue of individual fault and failure, but rather the failure of society to ensure that adequate systems, funding and support are in place so that all people, even in crisis situations, have access to housing.” The quotes commemorate the pivotal role of consumer/survivor voices in the public debate about effective population health strategies. Since then, consumer/survivor voices have proved that negative treatment experiences and stigma still constitute major reasons for limited service use in many countries [31,32,33,34,35]. The narratives collected by Moore-Nadler, Clanton, and Roussel, for example, made universal mechanisms in the interaction patterns between HSU and services transparent [33]. Starting from the social determinants of health, the peer voices uncovered how compromised systems, professionalism, and dehumanization affect engagement and funnel citizens into downward trajectories. Health equity is not served.

This paper highlights the functioning of the Dutch homelessness strategy through an intersectoral, integrating, recovery-oriented Quintuple Aim lens. The Dutch healthcare and welfare system is extensive. Separate services provide cures and care [36]. Highly specialized services with extensive in- and exclusion criteria are typical. Ambulatory outreach for people with SMI is provided by F-ACT teams; this is the de-facto mental health standard in the Netherlands [37]. Work support, welfare benefits, and homeless services are part of the social domain, services by a different infrastructure. Municipalities are responsible for service access, and for the connections between social and health services provided by local service networks. 

From 2006 to 2014, the “National Strategy Plan for Social Relief” was operational [38]. This national homelessness strategy intended to end the nuisance giving street homelessness in the four biggest cities. A multisite cohort evaluation confirmed that the program realized housing stability [39]. A secondary review nuanced its effects on the sustainability of the recovery results [40]. From 2015 on, the Dutch government implemented austerity measures to safeguard the social security system. The responsibilities of municipalities for disadvantaged and disabled citizens increased, assuming that local parties are better placed to coordinate care. At that time, people without an address fell off the governmental radar. They were often excluded from monitoring and underrepresented in social surveys. The Homeless People Treatment and Recovery (HOP-TR) study was conducted to meet the information needs of municipalities and homeless services. The systematic screening of symptomatic health revealed the burden of health problems and the interdependent character of the HSU’ needs [41]. Three out of four (72.5%) needed dynamic, intensive care because of SMI. By contrast, the estimated prevalence of SMI in 2013 in the adult Dutch population was 1.7%. Therefore, the results exposed extreme disparities. This paper aims to use the HOP-TR data to expand the understanding of where and why health inequity occurs in the treatment of marginalized populations deprived of a home. It uses two questions to focus on the responsiveness of local services to HSU’s needs:What are the main patterns of vital needs in the HSU population?How do local services respond to the main patterns of vital needs?

## 2. Materials and Methods

### 2.1. Study Design and Recruitment

A separate paper has documented the methods used in the HOP-TR study [42]. In seven Dutch cities, local health reviews were conducted in various types of homeless facilities. A naturalistic meta-snowball sampling resulted in a representative sample of homeless services (N = 16) and HSU (N = 436) between 2015–2017. Semi-structured interviews of HSU collected health and needs data. Personal biographies and standardized instruments were included, such as the interRAI Community Mental Health questionnaire (CMH) [43]. A comprehensive, rights-based strategy allowed network assessments from the perspective of HSU and professionals. 

The HOP-TR strategy assessed health and needs in an ecological vulnerability–stress approach. This considers marginalization and recovery a continuum. It structured data in the symptomatic, social, and personal dimensions of recovery. Integrating clinical assessments were used to describe the presence of physical health problems, transdiagnostic mental health (MH) features, and MH-related care needs. An analytical epidemiological approach exposed the burden and interrelationships of Mental Illness (MI), substance use, intellectual impairments, and physical health problems [41].

The decision tree in Figure 1 defined the assessments of Mental Health-Related Care Needs (MHRCN). It is based on the Dutch consensus EPA (acronym for SMI) [44]. 

Using this decision tree, the sample was divided into subgroups with different levels of MHRCN: No Needs (NN: 5.3%), Conditional Needs in relation to MI (CN: 22.3%), and Intensive Needs in relation to SMI (IN: 72.5%) [41]. Subjects were categorized in the NN-subgroup if they had no MH vulnerabilities or displayed sufficient personal resilience and social support to overcome the problems observed. The pervasive nature and the circular interrelations between symptoms and disabilities were key criteria to discriminate between (temporary) conditional and (long-term, dynamic) intensive needs. 

Recruitment aimed to add settings as long as additional reviews generated new insights into health patterns. An analysis of variance was conducted, based on the subsample means of health problems, and MHRCN. The methods paper proves that subsample addition was continued until topic saturation was reached [42]. 

### 2.2. Variables and Statistical Analyses

This paper highlights the responsiveness of local services to essential needs of HSU for the enjoyment of health, full citizenship, and recovery. Maslow’s hierarchy of needs provides a relevant framework for structuring needs in light of growth [45]. Its foundation consists of vital basic and health needs, targeting essential physiological and safety conditions for daily functioning and participation in society. Therefore, “vital needs” are operationalized as HSU’s basic and health needs in Maslow’s hierarchy. 

Various individual and contextual factors can result in marginalization or recovery, such as personal strengths, vulnerabilities, social resources, caregiver relations, and interactions with services. We used an analytical epidemiological approach to assess vital needs and the responsiveness of local services. Acknowledging the divergent character of the MH needs, we focused on the total sample and explored the differences between the three subgroups. 

Previous research had identified the high prevalence of health problems in the HSU population [41]. The sample characteristics of HSU biases vital needs HSU to a home, an income, and education. In the exploration of meaningful activities, most HSU expressed the need for paid work. 

Focusing on question one, we used three steps to identify main patterns of vital needs. Step one defines needs domains from the vital needs in the HSU’ perspective. In the Netherlands, high-level administration skills are fundamental to navigating complex bureaucracies for the fulfillment of vital needs. Administration needs define support needs to overcome functional illiteracy. Paid work needs reflect lacking certifications or skills for sustainable chances to acquire and hold a regular job. Further, physical needs reflect regular monitoring needs because of chronic physical conditions, including consequences of sex work and hard drug use. MH needs were operationalized as the burden of MH vulnerabilities (such as substance use, intellectual impairments, and/or other transdiagnostic features of MI). We also compared the impact of trauma in daily life, self-harm indicators, and violence symptoms in the subgroups.

Step two defines the co-occurrence of needs domains as cumulative needs. We calculated the sum scores, and the subgroup means of separate needs components. The need for a home was 100% since all subjects were homeless according to the European Typology and Housing Exclusion (ETHOS) [46].

Finally, step three considers the patterns of concurrent needs. We computed the presence of the various sets of vital needs. We created no/yes variables to count the presence of needs in four domains: paid work, administration, physical, and mental health. With the same no/yes assessments we computed how many needs domains were involved. The category “single needs”, for instance, denotes that needs were present in only one domain, while needs in the remaining three domains were absent. 

For question two, we used three steps to evaluate the responsiveness of local services to vital needs. Step one focused on the presence and deliverables of social, and health services. Basic interview data were recoded to discriminate between structural MH service contacts and incidental consultations such as intakes or care guidance. 

Step two evaluated delivered services in relation to present individual needs. Therefore, we used a strategy, inspired by the Camberwell Assessment of Needs (CAN) [47], with the value labels “no need”, “serviced needs”, and “unserviced needs”. Assessments were made irrespective of the success of the support. Services responded to administration needs if HSU with functional illiteracy received administrative support. Paid work needs were serviced if the support of HSU without a job was targeted at regular work. Serviced physical needs indicate that subjects with physical needs at least once visited a physician in the last 3 months. Collected data about emergency room visits and overnight hospital stays were not included, since the assessments concentrated on preventive monitoring from the perspective of recovery. Further, the appraisal of the service responsiveness to MHRCN was guided by the consensus of SMI/EPA. In the CN-subgroup, any MH service contact in the last year resulted in “serviced needs”. In the IN-subgroup, both the engagement of a multidisciplinary team and MH service contact in the last week were required for computing “serviced needs”. 

Finally, step three examined the “serviced needs” responses in the presence of concurrent needs. The tables represent the percentages of the subgroups and the total sample. The variables overview in Appendix A depicts the source and operationalization of the variables represented in Table 1, Table 2, Table 3 and Table 4. Except for age and some quotes, all variables are categorical. “Could not/would not respond” answers were recoded as missing values. All missing values were below 2.0%. Chi-square tests were run to compare the subgroups. 

## 3. Results

### 3.1. Sample Profile

Table 1 describes the background characteristics of the sample.

Most HSU are persistently or intermittently homeless. Even subjects living independently were at risk of losing their homes. Regularly, traumatic childhood events or migration disturbed normal school and working careers. Most are low-educated (82.3%) males (81.0%) with a migration background (52.1%). Only 2.8% actually have a regular job. Some never had a paid job in the Dutch labor market (14.0%). Some recently lost their jobs and were currently unemployed (4.8%). The vast majority were persistently unemployed (55.5%). Financial problems were common. Two out of three report trade-offs (65.8%). Most were unable to size up the outstanding debts. The interviews revealed that financial uncertainties limited care access. 

### 3.2. Main Patterns of Vital Needs

Table 2 and Figure 2 reflect the main patterns of vital needs. Table 2 starts with the individual needs (step one). Figure 2 represents the cumulative needs in the subgroups (step two). Then, Table 2 continues with the concurrent patterns of vital needs (step three). 

#### 3.2.1. Individual Needs

Table 2 starts from the basic needs because of the key role of municipalities in local service networks and the social domain. All HSU needed a home, mostly an independent place to live (92.7%). Only some need assisted or 24 h supervised accommodation, due to severe physical, neurocognitive, or intellectual impairments. In addition, paid work needs are common (82.3%). Over half the HSU have administration support needs (57.7%).

Symptomatic health patterns were described previously [41]. The increasing burden of MH problems with the MHRCN confirms that the decision tree successfully differentiated between No, Conditional, and Intensive MH-Related Care Needs. While the interviews revealed similar subgroup results regarding the early onset and the high burden of traumas, the impact of previous trauma differs significantly. In the IN-subgroup, they produce daily life symptoms such as immediate safety concerns in four out of ten. As represented, more expressions of violent behavior and self-harm indicators also demonstrate the higher burden of MH needs in this subgroup. 

#### 3.2.2. Cumulative Needs

The results revealed that almost all HSU were bothered with financial problems or income uncertainties. Therefore, in the cumulative needs, the need for an income was stated at 100%. Figure 2a presents the mean cumulative needs of the MHRCN subgroups. The value labels represent the means of the single needs components. Figure 2b displays the distribution of the cumulative needs of each subgroup. 

As shown, the mean MH needs differ significantly. Consequently, the subgroup means of cumulative needs also show highly significant differences (*p*: 0.000). The differences in the means of physical and basic needs were not significant.

In the NN-subgroup (*n* = 23), the mean cumulative needs add up to 5.9 (SD 2.0, range 2–9). The basic needs are dominant. As expected, the burden of health vulnerabilities is low. Anxiety, depression, and trauma were most common, while over-/underweight and cardiovascular problems resulted in physical monitoring needs. 

In the CN-subgroup, the mean cumulative needs count is 9.0 (SD 2.5, range 3–18). The ranking of MH problems was headed by addiction, anxiety, trauma, and personality problems. By far, over-/underweight and hard drug use were the most common physical needs. 

In the IN-subgroup, mean cumulative needs sum up to 10.6 (SD 2.6, range 4–20). Still, MH problems were headed by addiction but closely followed by anxiety, trauma, agitation, and depression. The physical monitoring needs related to hard drug use, accompanied by comorbidities such as over-/underweight, and gastrointestinal, neurological, lung, and cardiovascular problems. Figure 2b exposes the substantively higher level of cumulative needs in the subgroup. In 90.8% of the subgroup, the cumulative needs vary from 7 to 14. The dynamic and intensive character of the subgroup needs is also set by the pervasive nature of SMI and the circular relations between symptoms and disabilities. 

#### 3.2.3. Concurrent Needs 

Table 2 shows that basic needs are rarely limited to an income and a home. Only one out of eight (15.8%) has no needs except an income and a home. Over half of the HSU have paid work and administration needs, in addition to the need for an income and a home (54.8%). The presence of health needs increases significantly with the MHRCN level. Consequently, most HSU had both physical and MH needs (69.3%). As shown, the needs of most HSU touch three or four domains (76.8%).

### 3.3. Responsiveness of Local Services to Main Patterns of Vital Needs

We assess the HSU’s perceptions with some quotes: “I need a place on my own. It’s difficult to live together with so many people.”“I don’t feel free. I’m afraid that I’ll lose control of myself and hurt somebody.” “I need a home and a paid job. I want to take care of my son.”“I need a bed, bread, and a dignified existence, despite my debts.”“I need a paid job and a home, so that I can break with my criminal activities.”“They let me down frequently. I’ll freak out if they turn me out to the street again.” “My life is hopeless. I’m belittled everywhere.”“It takes much too long. I can notice that I’m irritable. I’m unable to carry on.”“I hate it here. They do nothing for me.” “Society pays more to keep me in the shelter than to get me out.”“I prefer sleeping rough. The care in the shelter does not suit to my needs.”

Many HSU were unable to reproduce their case managers’ names. They felt disappointed about treatment and service quality: services and society had let them down. Table 3 and Table 4 cover the efficiency of care and allow an appraisal if the HSU were right. Table 3 portrays the presence and deliverables provided by local services. Table 4 represents the responsiveness of local services to separate and concurrent vital needs. 

#### 3.3.1. Service Presence and Deliverables

Table 3 reports the contacts of HSU with local services over the past six months.

The results show differences in the service deliverance in the three subgroups. HSU with intensive needs less frequently receive support aimed at obtaining a paid job. They have significantly more contact with ambulant MH services, as expected due to the intensive character of their needs. Similarly, consistent contacts of outreach services and scheduled MH therapies are also targeted to this subgroup. Physician visits show a decreasing trend with the increase in MHRCN.

#### 3.3.2. Responsiveness to Single Needs 

Table 4 considers the serviced services when needs were present. It shows how provided services relate to the presence of separate needs. As shown, almost one out of two HSU with administration support needs actually received care. This compares favorably with the responsiveness to paid work needs. If MHRCN were *absent*, present paid work needs were serviced in one out of five. In the IN-subgroup, serviced needs were below one out of ten (*p* = 0.000). The work participation rate among HSU without certifications was 2.5%; it increased to 3.9% if certifications were sufficient. 

Similarly, the physician contacts were inadequate to respond to the physical needs. In the NN-subgroup, one out of three HSU at least once visited a physician in the last months. Despite the increasing burden of health monitoring needs, the proportion worsened to one out of five in the IN-subgroup. 

The responsiveness to single basic and physical health needs is low, but the responsiveness to the MH needs is even lower. As shown, in the CN-subgroup, only 4.1% had appropriate contact with MH services. If the presence of IN, the MH service contacts were appropriate in 8.9%.

#### 3.3.3. Responsiveness to Main Patterns of Vital Needs 

Table 2 shows that most HSU have both mental and physical health needs (69.3%). Additionally, a lot have paid work and administration support needs (54.8%). Table 4 shows how received services relate to the presence of concurrent needs. Only 2.3% of the HSU with mental and physical health needs actually receive appropriate mental and general healthcare. Similarly, administration support and paid work needs are actually met in 3.3%. Further, the results demonstrate that concurrent needs are hardly ever met if they exceeded two domains. 

## 4. Discussion

Healthcare and social services aim to ensure health equity for all users. Despite ongoing efforts, marginalized populations remain underserved. This paper describes Dutch homelessness care. Empirical data, collected in a representative sample of HSU, focus on vital needs. This paper evaluates the responsiveness of local service networks to citizens’ vital needs through an intersectoral, integrating, recovery-oriented lens. 

The main study results show that most HSU have both mental and physical health needs (69.3%). Similarly, many need administration and paid work support (54.8%). Most people have multiple needs (three or four domains: 76.8%). Even in the absence of MHRCN, the needs are seldom limited to income and a home (the default domains in this sample). 

By contrast, the results demonstrate the inadequate responsiveness of local services to main patterns of vital needs. Considering needs in two domains, coverage by care ranged only from 0.6–13.1%. Concurrent needs over two domains were hardly met. 

Previous research uncovered the intensive, integrating care needs in relation to SMI. Additionally, this paper discloses the high, multidimensional needs in the CN-subgroup. The divergent patterns prove that all Dutch HSU need integrating care. Furthermore, the poor responsiveness of service networks demonstrates the universal failure to cover vital needs in marginalized populations with interdependent needs. 

Thus, the data show underserving and uncover some causes of health disparities. First, the responsiveness to the vital needs of the subgroups shows significant differences. In the CN-subgroup, present MH needs were appropriately met in less than one out of twenty (4.1%). In the IN-subgroup they were met in less than one out of ten (8.9%). The results ascertain that the huge majority is underserved. Moreover, the increase in ambulant MH contacts is proportionally low in relation to the higher burden of health needs in the IN-subgroup (Table 2, Figure 2b). Underserving further increases in an ecological perspective over time. This explains the higher prevalence of previous homelessness in the IN-subgroup. Further, the substantially higher expressions of violence and self-harm indicators are both causes and consequences of the marginal living conditions. 

Second, recalling the low work participation (2.8%), only few HSU with paid work needs actually received paid work support (11.1%). In the absence of sufficient certifications, the work participation rate was extremely low (2.5%), and it hardly increased if certifications were present (3.9%). Corresponding figures of the general population show a work participation rate of 67.2% [48]. The workforce participation among adults without essential certifications was 50.5% and increased to 74.3% for persons with certifications. Thus, work support is neglected in HSU care. While the handicaps can explain low employment (factor 20), one would expect at least attempts from services. 

Finally, the data prove the failure of local service networks to meet integrating care needs. Higher complexity relates to a higher level of unserviced needs. The vital character of the needs, in Maslow’s hierarchy, makes this failure dramatic. Can it, in the care for marginalized populations, be accepted that services respond to less than one out of two present needs? The HSU morbidity and mortality figures reveal inequity [22,23,41]. This data show that the current Dutch homelessness care strategy will not improve the health, and morbidity and mortality rates.

Moreover, the paper provides insight where and why health inequity occurs in the treatment of marginalized populations deprived of a home. Certified human rights treaties advocate the highest standard of health for persons with disabilities [49]. The harsh reality in daily life of HSU uncovers that even vital minimum standards remain unmet. The data show that even universal health coverage in high income countries does not limit health inequity. The “where” is in line with the literature, showing that homelessness in high income countries, with low poverty levels, mainly affects populations with interdependent needs [50]. The data offers a modest view on numerous reasons why things in HSU—service interactions continuously go wrong [25,26,30,31,32,33,34,35]. In the light of ecological process-oriented frameworks [5,24], the data underline that needs insufficiently lead to care, resulting in inadequate service access and engagement. The data match the universal mechanisms described by Moore-Nadler and colleagues [33].

More importantly, the consistent failure to meet integrating needs exposes the costs of overspecialized, separate, specialized healthcare and social service systems [51]. It suggests that in local service networks the linear approach of recovery is still the norm. As long as the responsiveness to supposed safety risks or health symptoms are dominant, more contacts with outreach services or scheduled MH therapies are no guarantee for long-term integrating recovery-oriented care. Instead, they may lead to over-emphasis on crisis management, lacking attention for the ecological, process-oriented approach to recovery [27]. The powerful role of professionals, the focus on restoring control, and the assumption of the malleability of growth are detrimental to relational aspects and the human need of autonomy [52,53,54]. Meanwhile, workers within services need to fulfill their tasks within complex frameworks of organizational instructions, professional norms, and funding. In addition, care for marginalized populations with interdependent needs constantly requires awareness of fundamental rights and personal and societal safety [29,49]. Homelessness is neither an individual fault, nor an individual workers’ issue. The long-term character of homelessness and the universal failure to safeguard vital needs uncovers the absence of comprehensive integrating strategies focusing on recovery. 

An important factor is how HSU perceive care. Many preferred sticking to their own survival strategies, as they were not satisfied with how their needs were serviced in the past. Many uttered feelings of indignation or disappointment: in their perception, services or society had let them down. In sum, the data show that the HSU were right. The shared decision for care choices was not realized. These results match the findings of the Canadian Observatory on Homelessness [30]. Backed with consumer/survivor narratives, the data reveal the unchanged, central role of power imbalances, negative treatment experiences, and stigma in causing and maintaining health inequity.

For services, developers, and policy-makers, these results address major societal challenges to transfer comprehensive health and social care knowledge from clinical guidelines to practice [55,56]. Important gaps in implementation knowledge were uncovered. How can we facilitate the implementation of successful strategies in different sociocultural contexts [16]? Should we prefer personal continuity or specialized care models [57,58]? What about the role of peers and Recovery Education Centers [59,60]? What will work out well? It is clear that targeting single ingredients (cultural sensitivity, trauma-responsive care, service leadership, etc.), will not suffice. Instead, substantial improvements require a radical commitment to all ingredients of the ecosystem of care. HSU urgently need comprehensive, integrating, socio-ecological ecosystem strategies that better serve the needs of marginalized populations with interdependent needs [27,42]. A crucial concept, defined in the quintuple aim care outcome model, is health equity. This will only be realized by continuously listening to user voices and exploring divergent strategies. Given the current Dutch service system, collaborative networks within neighborhoods, cities, and regions are in charge to initiate change. The data offer crucial insights that help citizens build collaborative networks with respect for personal autonomy and the ability to shape reinforcing conditions at all ecosystem levels for sustainable recovery over time. 

Limitations—All data were collected in single assessments by a single interviewer. The calculation of the responsiveness rates was based on the presence and deliverances of local services in relation to present needs, but do not allow assessment of successful recovery strategies in time. 

Single-person assessments might induce a bias but offer the best health estimates of a hidden part of the Dutch HS population. The data quality is limited to information collected during single face-to-face encounters. Better care access and additional check-ups certainly would provide more reliable descriptive health data. 

The HOP-TR database was built over different local reviews, commissioned by municipalities and service providers. As expected in this skewed sample of the Dutch inhabitants, the vital basic needs show similar patterns in all subsamples. A variance analysis based on health patterns disclosed that subsample addition was continued until topic saturation was reached. Because the data collection concentrates on HS users in traditional homeless services, population selection bias might limit the generalizability of the results. Evidently, homeless people with insecure or in inadequate housing, such as “sofa surfers” and “work migrants lodging in holiday parks”, are underrepresented. Further, the cross-sectional design overrepresented individuals with complex health problems by assessing the most needing individuals who stay in the facilities for a long time. Considering the hidden nature of this population, the HOP-TR recruitment strategy is the best possible approach to comprehensively assess a representative sample of Dutch adult HS users in 2015–2017.

The HOP-TR data were collected before the COVID-19 pandemic. A Dutch national research project monitored the morbidity, mortality, policy, and healthcare use among HSU during the pandemic [61]. HSU was identified as a high-risk group, but targeted policy measures appeared effective to manage the risks. The incidence of COVID-19 among Dutch HSU exactly followed the general population trends. As expected, the hospitalization rate of identified COVID-19 cases was higher among HSU than figures from the general population (10% versus 4%). This can be explained by the higher burden of chronic health conditions among HSU. Temporary municipal policy measures in response to the pandemic did not resolve the shortages of affordable housing or the underlying causes of homelessness in the Netherlands. Therefore, there is no reason to expect that preexistent, extreme disparities and health inequity since the pandemic disappeared. 

Strengths—The study provides accurate intersectoral data of an invisible part of the Dutch inhabitants, disclosing where and why disparities exist. Data are collected in face-to-face interviews with a trained clinician.

## 5. Conclusions

This paper presents data on Dutch homeless service users and provides insight into where and why health inequity occurs in the care of marginalized populations deprived of a home. A comprehensive, recovery-oriented approach collected health and needs data from service users’ and professional perspectives. Starting from the users’ vital needs, we evaluate the responsiveness of local services to the main patterns of vital needs. The data disclose the failure of healthcare and social service systems to meet integrating care needs over various domains of needs. 

The paper uses a dynamic process approach to unravel what happens in the interactions between services and users. It builds on and contributes to the socio-ecological literature on service access, by making it transparent that universal health coverage in high-income countries does not prevent the occurrence of health inequity. A modern care system aiming for the highest attainable standard of health failed to provide minimum standards to meet vital needs in marginalized populations. It warns that system failure can occur in extensive (over-specialized) service systems.

These results denote the challenge of local service collaboratives to develop robust strategies focusing on essential conditions for recovery. Rights-based ecosystem approaches incite attention to different voices and power balances in user—service interactions. Such strategies allow divergent care strategies and enable normalization and growth in a sustainable way. Embedding the quintuple aim in recovery-oriented care strategies may improve local service networks and keeps the focus on health equity.

## Figures and Tables

**Figure 1 ijerph-20-02546-f001:**
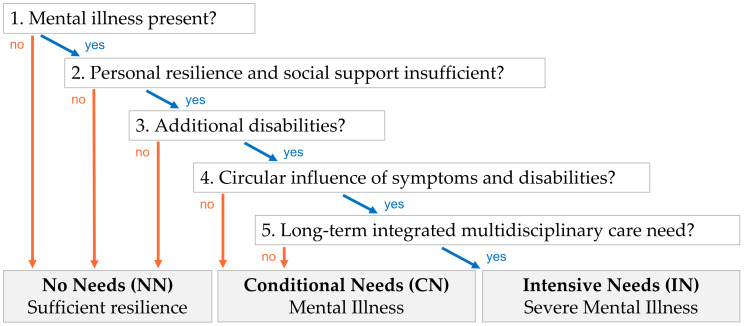
Mental Health-Related Care Needs assessment.

**Figure 2 ijerph-20-02546-f002:**
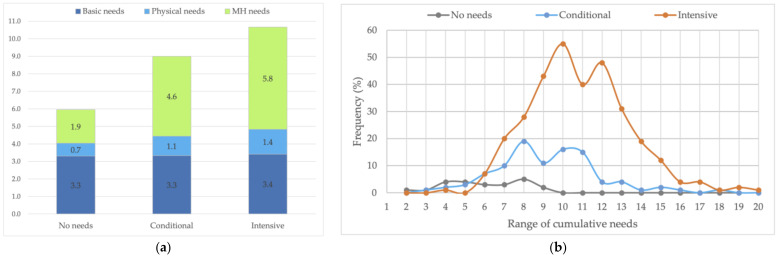
Cumulative needs by subgroup. (**a**) Means and composition. (**b**) Frequency distribution.

**Table 1 ijerph-20-02546-t001:** Background characteristics of subgroups and total sample.

		*Mental Health-Related Care Needs*				
		*No Needs*	*Conditional*	*Intensive*	*Total*	
		*n* = 23	*n* = 97	*n* = 316	*n* = 436	
Homelessness history	Roofless	78.3	77.3	72.3	74.1	
	Houseless	21.7	22.7	27.2	25.9	
	<3 months	13.0	14.4	11.7	12.4	
	3–12 months	26.1	30.9	19.0	22.0	
	>12 months	60.9	53.6	69.3	65.4	
	Previously homeless	60.9	72.2	82.0	78.7	*
Age	Mean (years)	46.4	43.7	42.4	42.9	
Gender	Male	73.9	80.4	81.6	81.0	
Migration background	Native	30.4	42.3	50.9	47.9	*
	Western migrants	13.0	9.3	14.2	13.1	
	Non-Western migrants	56.5	48.5	34.8	39.0	
Education level	Low	69.6	80.4	83.9	82.3	**
	Middle	13.0	18.6	13.9	14.9	
	High	17.4	1.0	2.2	2.8	
Work status	Unemployed	73.9	62.9	77.9	74.3	**
	Temporary supported work	8.7	20.6	19.0	18.8	
	Regularly employed	4.3	8.2	0.9	2.8	
	Retired, student	13.0	8.2	2.2	4.1	
Income status	No income	13.0	10.3	16.1	14.0	
	Benefit inquiry submitted	21.7	15.5	13.6	14.4	
	Benefit inquiry granted	65.2	74.2	70.3	70.9	
	Trade-offs	43.5	67.0	67.1	65.8	

Except age, all figures are percentages. * *p* < 0.05 ** *p* < 0.01.

**Table 2 ijerph-20-02546-t002:** Focus of vital basic and health needs in subgroups and total sample.

		*Mental Health-Related Care Needs*				
		*No Needs*	*Conditional*	*Intensive*	*Total*	
		*n* = 23	*n* = 97	*n* = 316	*n* = 436	
Basic needs	Independent living	100.0	99.0	90.2	92.7	*
	Assisted living	0.0	0.0	5.4	3.9	
	Hospital or 24 h supervised	0.0	0.0	2.8	2.1	
	Paid work	69.6	80.4	83.9	82.3	
	Administration	60.9	52.6	57.6	56.7	
Health needs	Physical monitoring needs	60.9	68.0	71.2	70.0	
	Mental health vulnerabilities	73.9	100.0	100.0	98.6	**
	Mental illness	60.9	94.8	99.1	96.1	**
	Substance use	21.7	73.2	83.5	78.0	**
	Intellectual impairments	0.0	36.1	44.0	39.9	**
	Trauma impact	13.0	21.7	42.4	36.2	**
	Self-harm indicators	4.3	7.2	23.4	18.8	**
	Expressions of violence	8.7	17.5	40.8	33.9	**
Concurrent basic needs	No needs	26.1	17.5	14.6	15.8	
	Only paid work	4.3	2.1	1.6	1.8	
	Only administration	13.0	29.9	27.8	27.5	
	Paid work and administration	56.5	50.5	56.0	54.8	
Concurrent health needs	No health needs	13.0	0.0	0.0	0.7	**
	Only physical	13.0	0.0	0.0	0.7	
	Only mental	26.1	32.0	28.8	29.4	
	Both mental and physical	47.8	68.0	71.2	69.3	
Divergence	Single needs	13.0	9.3	6.0	7.1	
	Double needs	26.1	15.5	15.2	15.8	
	Triple needs	26.1	40.2	38.9	38.5	
	Quadruple needs	30.4	35.1	39.9	38.3	

Percentages. * *p* < 0.05 ** *p* < 0.01.

**Table 3 ijerph-20-02546-t003:** Service presence and deliverables.

		*Mental Health-Related Care Needs*				
		*No Needs*	*Conditional*	*Intensive*	*Total*	
		*n* = 23	*n* = 97	*n* = 316	*n* = 436	
Social services	Paid work support	30.4	27.8	11.1	15.8	**
	Administrative support	39.1	50.5	42.7	44.3	
General health services	Any physician visits	26.1	21.6	20.3	20.9	
Mental health services	No ambulant program	100.0	95.9	62.0	71.6	
	Monodisciplinary care	0.0	4.1	15.8	12.4	**
	Multidisciplinary care	0.0	0.0	15.5	11.2	
	Contact in last year	0.0	4.1	38.0	28.4	**
	Contact in last week	4.3	0.0	18.0	13.3	**
	Intended MH treatment	0.0	1.0	8.7	6.7	*

Percentages. * *p* < 0.05 ** *p* < 0.01.

**Table 4 ijerph-20-02546-t004:** Focus of delivered services in relation to present needs.

			*Mental Health-Related Care Needs*				
			*No Needs*	*Conditional*	*Intensive*	*Total*	
			*n* = 23	*n* = 97	*n* = 316	*n* = 436	
Paid work			18.8	17.9	8.7	11.1	**
Administration			42.9	54.9	43.4	45.7	
Physical health			35.7	25.8	21.8	23.3	
Mental health			0.0	4.1	8.9	7.4	**
Mental health	Physical health		0.0	1.5	2.7	2.3	
Mental health	Paid work		0.0	0.0	0.8	0.6	
Mental health	Administration		0.0	3.9	2.2	2.5	
Physical health	Paid work		0.0	3.6	2.1	2.3	
Physical health	Administration		22.2	16.7	11.5	13.1	
Paid work	Administration		7.7	6.1	2.3	3.3	
Mental health	Physical health	Paid work	0.0	0.0	0.5	0.4	
Mental health	Physical health	Administration	0.0	2.8	0.0	0.6	
Mental health	Paid work	Administration	0.0	0.0	0.0	0.0	
Physical health	Paid work	Administration	0.0	0.0	0.8	0.0	
Health	Paid work	Administration	0.0	0.0	0.0	0.0	

Percentages. * *p* < 0.05 ** *p* < 0.01.

## Data Availability

The first author is using the data of the HOP-TR study to make a PhD. When the PhD is finished, the data will become available in a public repository. Until then, we are open to data requests within the scope of collaborative projects.

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
