# Peer review of "Vital Needs of Dutch Homeless Service Users: Responsiveness of Local Services in the Light of Health Equity"

_ijerph, 2023, doi:10.3390/ijerph20032546_

Round 1

Reviewer 1 Report

Using the Dutch HOP-TR study data, this paper described the characteristics of the homeless service users, and found that their vital needs are largely unmet. I have the following comments for this study:

1.     In introduction section, authors said “This paper aims to use the HOP-TR data to expand the understanding where and why health inequity occurs in the treatment of marginalized populations deprived of a home. I think the “why” part needs more discussion. I agree that authors found evidence that homeless individuals’ vital needs were largely unmet. However, is it due to the growing number and need from homeless individuals? Or lack of supply from the corresponding social service programs (if yes, is the shortage in financial resources, human resources, institutional/policy resources, or others)? Or there are issues (e.g. inefficiency) in implementation or management? More elaboration on these issues would be helpful to answer the “why” question.

2.     In methods section author said “A naturalistic meta-snowball sampling resulted in a representative sample of homeless services”. I am concerned with the potential sample selection bias from snow ball sampling. I think more explanation and justification is need to argue that this sampling is “representative”.

3.     Authors used data during 2015-2017, which were 6 years old. Is newer data, especially since the COVID-19 pandemic, available? If not, more explanation (and maybe comparison) is needed so we feel relatively confident that the results from this data are largely representative and applicable to nowadays.

4.     Authors only documented percentage results in the 4 tables. I suggest showing the numbers as well so readers could have a sense of both number and prevalence without calculating on their own. In fact, I am bit concerned with the much smaller sample size for the no need group and conditional group. Are they potentially underrepresented, compare to the intensive group?

5.     Minor comments:

·       Authors used term “MI” and “SMI” in the manuscript, while used term “conditional” and “intensive” in the tables. Standardize the terminology/acronym would make the paper more reader friendly.  

·       Formatting: The fonts have been switching between Times New Roman and Calibri throughout the paper.

Author Response

See ijerph_2156842_revision1_reply_R1

Reviewer 2 Report

In the introductory part of the paper  more theory of health equity in relation to marginalized population would significantly enrich the study. In the discussion part of the manuscript, it is worth comparing the situation with our countries around the world. Conclusion part of the paper needs to be extended.

Author Response

See ijerph_2156842_revision1_reply_R2

Reviewer 3 Report

Thank you for the intresting manuscript.

In introduction you can mention inadequate access to healthcare described here ( https://brieflands.com/articles/jjcdc-114501.html)

Please describe methods in a separate chapter, including section participants. It will be easier to understand and read.

7 cities are all in one country? If yes, please in the title add on the end -" In the Netherlands "

Please change the writing style font (there is a different style of technical font) Not unique.

Author Response

See ijerph_2156842_revision1_R3

Round 2

Reviewer 1 Report

Authors' revision has addressed most of my comments. I don't have any additional comments.